# Continuous Glucose Monitoring System Based on Percutaneous Microneedle Array

**DOI:** 10.3390/mi13030478

**Published:** 2022-03-20

**Authors:** Ming-Nan Chien, Yu-Jen Chen, Chin-Han Bai, Jung-Tung Huang

**Affiliations:** 1Division of Endocrinology and Metabolism, Department of Internal Medicine, MacKay Memorial Hospital and Mackay Medical College, National Taipei University of Technology, No. 1, Sec. 3, Zhongxiao E. Rd. Da’an Dist., Taipei City 106, Taiwan; chienmingnan@gmail.com; 2National Taipei University of Technology, No. 1, Sec. 3, Zhongxiao E. Rd. Da’an Dist., Taipei City 106, Taiwan; allenbull54@gmail.com (Y.-J.C.); hank0530pt@gmail.com (C.-H.B.)

**Keywords:** biosensor, blood glucose, tissue interstitial fluid, continuous glucose sensing, microneedle, micro transfer

## Abstract

A continuous blood glucose monitoring system (CGMS) which include a microneedle-array blood glucose sensor, a circuit module, and a transmission module placed in a wearable device is developed in this research. When in use, the wearable device is attached to the human body with the microneedle array inserted under the skin for continuous blood glucose sensing, and the measured signals are transmitted wirelessly to a mobile phone or computer for analysis. The purpose of this study is to replace the conventionally used method of puncture for blood collection and test strips are used to measure the blood glucose signals. The microneedle sensor of this CGMS uses a 1 mm length needle in a 3 mm × 3 mm microneedle array for percutaneous minimally invasive blood glucose measurement. This size of microneedle does not cause bleeding damage to the body when used. The microneedle sensor is placed under the skin and their solutions are discussed. The blood glucose sensor measured the in vitro simulant fluid with a glucose concentration range of 50~400 mg/dL. In addition, a micro-transfer method is developed to accurately deposit the enzyme onto the tip of the microneedle, after which cyclic voltammetry (CV) is used to measure the glucose simulation solution to verify whether the difference in the amount of enzyme on each microneedle is less than 10%. Finally, various experiments and analyses are carried out to reduce the size of the device, test effective durability (approximately 7 days), and the feasibility of minimally invasive CGMS is evaluated by tests on two persons.

## 1. Introduction

According to the International Diabetes Federation, the number of diabetics worldwide will reach 552 million by 2030 [1]. High blood glucose levels in diabetes increase oxidative stress in the body, leading to various pathologies and complications; therefore it is important to monitor blood glucose strictly in patients with diabetes [2]. Most diabetic patients are troubled by regular blood glucose monitoring and only take intermittent measurements, which puts them at high risk of heart disease, blindness, and kidney disease [3]. How to reduce the discomfort and side effects of blood collection while taking into account the accuracy of sensing is an important issue for researchers [4].

Most of the commercially available physiological testing devices or tissue sampling by medical personnel require piercing the stratum corneum with a needle to extract tissue fluid/blood for analysis. Another important factor is that a large number of microorganisms on the surface of the skin can enter the body easily and cause infection, and damage to the glucose oxidase coating results in abnormal test values. Shaoguang Li’s team used hydrogel to coat the enzyme [5], which was shown to protect the enzyme layer and prolong the effective time of detection in an in vivo experiment on rats.

Many experts and researchers have attempted to develop non-invasive glucose meters. Caduff’s team used a non-invasive non-optical continuous glucose monitoring system to observe glucose changes in human tissue fluids [6] and Lahdesmaki’s team used microelectromechanical processes (MEMS) to integrate triple electrodes into contact lenses to measure glucose in tears [7]. However, at present, non-invasive blood glucose sensing still has problems such as sensitivity and background noise signals that need to be overcome, and the use of protective coatings such as poly HEMA may enable minimally invasive and accurate measurement of blood glucose changes [8,9].

Therefore, the main goal of this study is to achieve accurate and effective blood glucose measurement in diabetic patients, and reduce pain and discomfort during the process, which could improve the quality of life. We design a microneedle array base electrochemical sensor, a glucose detection circuit module, and a transmission module to be placed in a wearable device that could continuously detect the glucose concentration in interstitial fluid (ISF) with low invasiveness and transmit data to a cell phone wirelessly, with accurate measurement of changes in glucose concentration. The minimally invasive wound after use is shown in Figure 1. In this study, we construct a system for the self-health management of diabetic patients through a user-friendly testing platform [10,11,12].

Commercially available blood glucose devices differ among invasive, non-invasive, single-use, and continuous measurements. Invasive blood glucose meters are more accurate than non-invasive ones, and continuous measurements can show the change in blood glucose more clearly [13,14]. According to ISO15197, the error range of blood glucose meters can be accepted up to ±15%, which means there is still a lot of room for the development of a new blood glucose meter.

Commercially available continuous glucose monitoring systems such as the Dexcom G5 Mobile CGM System, FreeStyle Navigator, Medtronic iPro 2, and Abbott FreeStyle Pro all have sensors, transducers, and receivers. Referring to Table 1 [15,16,17,18], we can see that the length of the microneedle is about 5 mm~13 mm, the piercing site is located in the abdomen or arm and a warm-up time is needed before use to avoid inaccurate blood glucose readings. In our CMGS, because the length of the microneedle is 1 mm, no blood vessels are damaged after penetration into the skin, and we can start measuring in a very short warm-up time (5~10 min). The activity of enzyme used in CGMS may decrease when it is used for a longer time or frequency, so calibration is required at different time points to ensure the accuracy of readings. The objectives of this study are to verify the enzyme activity and durability of the sensor and conduct experiments to test the accuracy of the device using the sensor and receiver. The data from the receiver could be transmitted wirelessly to the computer or cell phone, making result reading easier.

## 2. Materials and Methods

The CGMS in this study can be divided into several components: the microneedle array sensor, the circuit board for conditioning the blood glucose sensing signal, the casing, and the wireless circuit module of Bluetooth MCU which transmits the signal to the cell phone. The block diagram of our CGMS is shown in Figure 2. The following is a description of the materials, important components, and the experiments that were conducted.

### 2.1. Micro-Transfer Method

In this study, the concept of microtransfer was applied to transfer the glucose oxidase onto the microneedle array sensor. This microneedle array was purchased from RichHealth Technology. The microneedle array sensor is shown in Figure 3a which includes three parts: a working electrode (WE), a counter electrode (CE), and a reference electrode (RE). The microneedle array is made of stainless steel SUS316L using a metal stamping fabrication process. Each microneedle is 1 mm in length, 0.25 mm width and 0.1 mm thickness. The counter electrode is then plated with gold, while the working electrode is plated with gold and PANI and the reference electrode is coated with Ag/AgCl. Each WE array has an area of 3 mm × 3 mm and is comprised of 3 × 4 microneedles. The working area of each microneedle array is about 1.2 mm^2^. All needles are beveled to prevent signal instability due to glucose oxidase detachment when piercing under the skin. A stamping device was used to transfer the glucose oxidase onto the surface of the microneedle array sensor, as shown in Figure 3b. The bevel of the microneedle should be taken into consideration when used for transfer. For this purpose, we designed a micro-transfer system that includes a dispensing machine, a stamping device, a fixture for the stamping device, and the pointing machine for transfer, as shown in Figure 3c,d, using fabrication parameters that could be controlled. 

The Lixi Technology Ltd.(New Taipei, Taiwan) 3-axis automatic dispensing machine which has a 3-axis moving working platform with a working area of 300 × 300 × 100 mm^3^ and a micro-stepping precision motor with a resolution of 0.001 mm/Axis and repeatability of ±0.02 mm/Axis was used in this study, as shown in Figure 3e. The final goal is to achieve mass production of enzyme coating with less than 10% error per piece [19]. Since there are 12 microneedles on a microneedle strip and each microneedle has a length of 1 mm and an inclination of about 70°, we require the ability to apply the glucose oxidase to every microneedle on the microneedle strip accurately and automatically.

The transfer process is shown in Figure 3f; a microneedle will be fixed to the positioning plate to ensure the correct position of the microtransfer. After positioning, the stamping device with the glucose oxidase is attached to the surface of the microneedle to hold the glucose oxidase in place; the total contact area is approximately 1.2 mm^2^ per contact. The stamping device needs a few seconds to replenish the glucose oxidase before it can continue to transfer the next microneedle.

### 2.2. Reagents and Immobilization

Glucose oxidase (GOD) is purchased from Tokyo industry Co., Ltd. (Tokyo, Japan) Phosphate, buffer saline (PBS. PH 7.2) and Nafion is purchased from Uni Onward. 2-Hydroxyethyl methacrylate (poly HEMA) is purchased from Sigma-Aldrich. Glucose (Dextrose anhydrous) is purchased from SHOWA chemical Co., Ltd. (Tokyo, Japan).

In this study we used the high polymer entrapment method, which means that after the glucose oxidase has been planted, the outer layer needs to be fixed with a high polymer coating. All layers of modified coating solution are as shown in Figure 4.

Au can enhance the strength of the PANI fixation, and the PANI layer is porous and therefore facilitates the fixation of enzymes [20]. Biocompatible insulating varnish reduces electrical conductivity in areas outside the glucose oxidase layer to ensure signal correctness. The GOD layer is to achieve specific detection of glucose [21]. The Nafion layer can improve adsorption of the object to be detected on the electrode surface and also mitigate interference from acetaminophen and uric acid. Nafion also helps exclude anion species which may be electroactive towards the electrode domain [22]. The poly HEMA layer is to protect the enzyme and blocks enzyme degradation from external substances to maintain the stability and durability of the electrode. The poly HEMA layer also acts as a glucose-limiting membrane.

### 2.3. Blood Glucose Sensing Circuit

This experiment uses L-type GOD to catalyze the reaction between blood glucose and oxygen to produce hydrogen peroxide, and then the electrode catalyzes the oxidation of hydrogen peroxide to produce electrons, and the blood glucose concentration can be calculated by the measurement of electric current. The chemical formula is shown in Figure 5.

The reference electrode (RE) is used to set the zero point so that current can flow through the working electrode (WE) and the counter electrode (CE), but the reference electrode will form a state of no current or almost no current. The current passing through the WE is measured and the relationship between the potential and current of the WE is measured simultaneously using these three electrodes. In this study it generates an oxidation reaction at the WE when glucose sensing is performed, which means that a positive charge is generated by the loss of electrons, and the current flows into the working electrode and then into the sensing circuit. In addition, the regulator establishes a reference voltage of 2.5 V for the circuit, thus providing a single power supply with very low static current consumption. Since the CE is a negative voltage (typically −300 mV to −400 mV) to the operating electrode, the amplifier draws enough current from the CE to maintain the control level between the WE and REF of the sensor. The RE is connected to the inverting input of the next amplifier, so there is no current flow. This means that the current flows from the WE and varies linearly with the glucose concentration. The transimpedance amplifier converts the sensor current into a voltage proportional to the glucose concentration and then uses the measured voltage to obtain the corresponding glucose concentration value to analyze whether valid discrimination is achieved. The test circuit diagram is shown in Figure 6.

The current concentration flowing into the working electrode pins in this circuit will be less than 100 nA, which is a very small current and therefore requires a trans-impedance amplifier (TIA) with a very low bias current to convert the current to an output voltage. The MAX9913 (purchased from Ltcore) operational amplifier is therefore well suited to this application because of its low bias current at room temperature and its high immunity to noise interference. Broadcom’s BCM20737S was selected as the Bluetooth transmission chip because it is a system-in-package (SiP) with a RF antenna and main frequency oscillator in a 6.5 × 6.5 × 1.2 mm footprint, saving the size of the oscillator and antenna matching. It also supports RSA4000 bit encryption and decryption to ensure secure data transmission.

During the study, the device was designed to fit into the sensing board, which already contains the Broadcom module BCM20737S, the sensing circuit based on MAX9913, and the microneedle sensor. The wearable device consisted of the top cover, the bottom casing, and the circuit carrier. The bottom casing has four contacts and slots to allow the microneedle sensor to be slid and placed in the slots. The sensing circuit board is connected to the microneedle sensor by four spring pogo-pins, and the circuit carrier is designed with springs on both sides. When this wearable device is placed to the human arm, and a layer of breathable tape or a tightening band is attached to the outside of the micro-device, a tightening pressure can be applied to the microneedle sensor on the assembled device to pierce the subcutaneous tissue, as shown in Figure 7.

## 3. Experimental Design

In this experiment, cyclic voltammetry was used to measure the characteristics of the microneedle and the constant potential meter used in this study is the CH instruments 1025B electrochemical analyzer. The electrode clamp was connected to the three selected electrodes and placed in the solution to be measured.

### 3.1. Micro-Transfer Enzyme Measurement on Microneedle Electrodes

To verify whether it is feasible to immobilize the enzyme on the microneedle electrode by transferring the enzyme, the parafilm on the microneedles was left uncovered, and glucose solutions with concentrations of 0 mg/dL, 50 mg/dL, 100 mg/dL, 200 mg/dL, 300 mg/dL, and 400 mg/dL were prepared the for the CV test. The results show the greatest change at 0.65 V (shown in Figure 8a) and a linearity in the data (shown in Figure 8b) at different concentrations, so we chose 0.65 V as the value to be tested.

#### 3.1.1. The Amount of GOD Used per Transfer

This experiment was carried out by comparing the signal size of enzymes from a 5 μL dispenser with that from a microtransfer to estimate the amount of GOD per transfer. The results are shown in Figure 9a. The difference in signal size allows us to estimate the volume of each transfer to be approximately 0.25 μL. To test this claim, a 0.25 μL enzyme was dispensed and compared to the microtransfer signal, the results of which are shown in Figure 9b, and the error was then collated from Table 2and verified to be less than 10%, which is within acceptable limits.

#### 3.1.2. Micro-Transfer Variations

In the variability test, a series of transfers were made for the parameters of time and the amount for replenishment of loss. Ten strips of working electrodes were transferred continuously, and four strips were sampled each time, and measurement of the coefficient of variation was carried out at 50 mg/dL and 400 mg/dL concentrations of glucose solution. The concentration of the solution, while the working electrodes were immersed, remained the same during the experiment. The coefficient of variation measurements of the four strips of working electrodes were taken at 0.65 V and the results are shown in Figure 10 and Table 3. The variability error was 5.1.24~5.220%, and the error was less than 10% of the industry standard for dispensing error.

### 3.2. Combed Test of Glucose Solution with Microneedle Array Sensor and Conditioning Circuit

To confirm the operation of the conditioning circuit and to understand its sensing range and sensitivity, testing for different concentrations of glucose solution in PBS: 0 mg/dL, 50 mg/dL, 100 mg/dL, 200 mg/dL, 300 mg/dL, and 400 mg/dL was carried out and the results are shown in Figure 11.

### 3.3. Blood Glucose Monitor Circuit Tested in an Agar-Based Skin Model

In this experiment, glucose solutions with concentrations of 50 mg/dL, 100 mg/dL, 200 mg/dL, 300 mg/dL, and 400 mg/dL were mixed in agar and covered with parafilm to simulate a subcutaneous medium [18] as shown in Figure 12a. A microneedle sensor and blood glucose monitor circuit were integrated as CGMS for the test by inserting the microneedle array into the agar as shown in Figure 12b. The results are shown in Figure 13a–e. From these figures, it can be seen that the continuous measurement of different concentrations of glucose solution was very stable during the four hours and the device combined with the blood glucose circuit and the microneedle sensor was very stable for detection on the agar matrix [23].

### 3.4. Long-Term Stability Test of Blood Glucose Circuit

Similarly, in the long-term stability test, we chose the above agar-based skin model for continuous monitoring with 100 mg/dL concentration of glucose solution to simulate subcutaneous tissue. The signals were read every five minutes for one week, with the microneedles of the CGMS penetrating the agar the entire time, and the results are shown in Figure 14. The enzyme concentration and blood glucose circuit system were stable during continuous monitoring and the microneedle sensor can be used for continuous monitoring in vitro for up to 7 days.

### 3.5. Human Experiment Test

This experiment was carried out on human subjects and the microneedle was measured externally before the test in order to calibrate the values, after which the microneedle and the circuit were fixed to the arm and the change in blood glucose after eating was recorded by a smartphone receiving a Bluetooth signal. From Figure 15a,b it can be seen that subject-1 started to measure blood glucose concentration and eat food at 13:45 p.m. and finished eating at around 14:20 p.m. Subject-2 started to measure blood glucose concentration and eat chocolate at 16:44 p.m. and finished eating at around 17:12 p.m. It can be seen that the blood glucose concentration increased slowly with the time after eating. The device, signal before and after meals, and the subcutaneous microneedle sensor were all stable and effective in distinguishing the levels of blood glucose concentration.

## 4. Conclusions and Prospects

The purpose of this study was to develop a wearable biomedical sensor module to be mounted on the human arm to achieve the goal of minimally invasive and long-term continuous blood glucose testing. At present, we have preliminary results on the device mechanism, microneedle sensor, blood glucose circuit system, and coating method. We have verified the feasibility of measuring blood glucose with this 1 mm needle length microneedle array, and the error range of the same product is within the relevant regulations. In the future, this kind of small minimally invasive wearable device is expected to be popularized and diversified, and this device can be used as a model to develop more kinds of analyte measurement, such as lactic acid, uric acid, cholesterol, adrenaline, etc.

## Figures and Tables

**Figure 1 micromachines-13-00478-f001:**
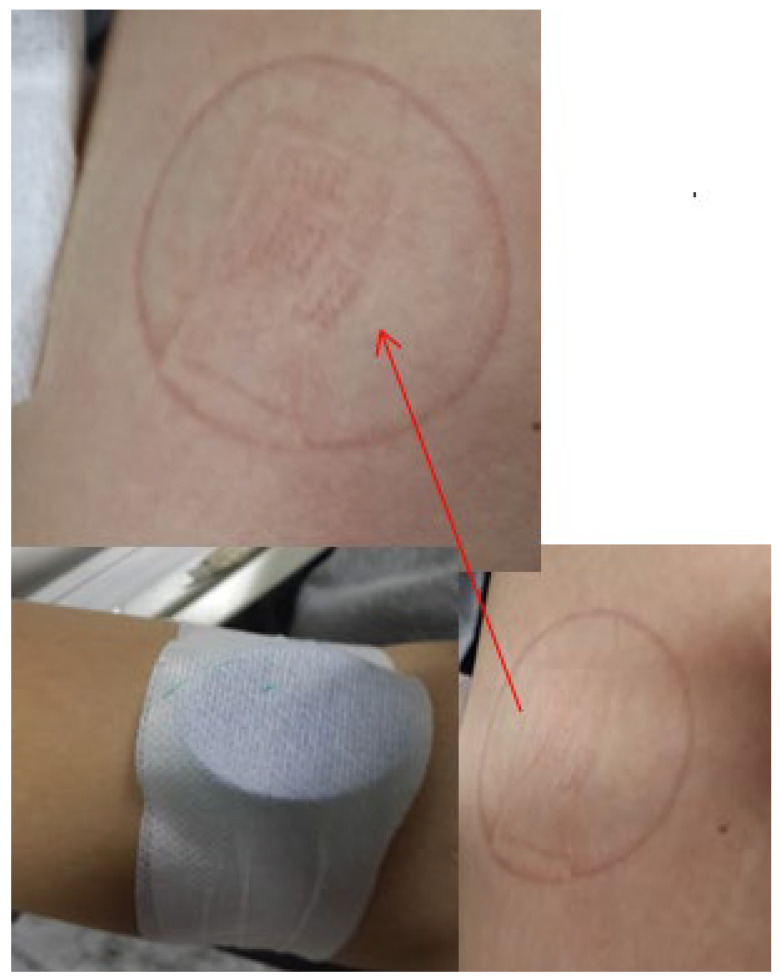
Minimally invasive wound after CGMS use, this small wound will cause almost no pain and no bleeding.

**Figure 2 micromachines-13-00478-f002:**
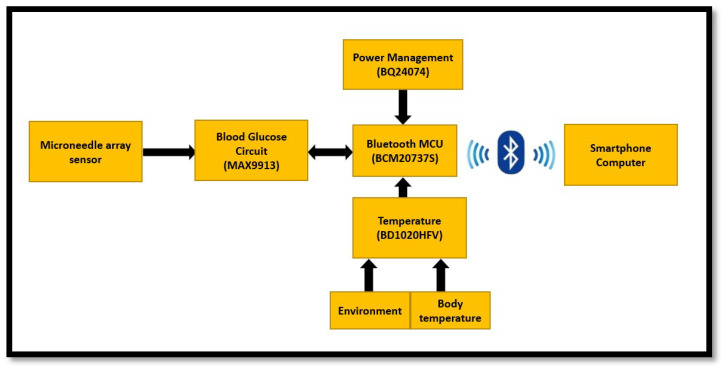
CGMS System Concept of Operation.

**Figure 3 micromachines-13-00478-f003:**
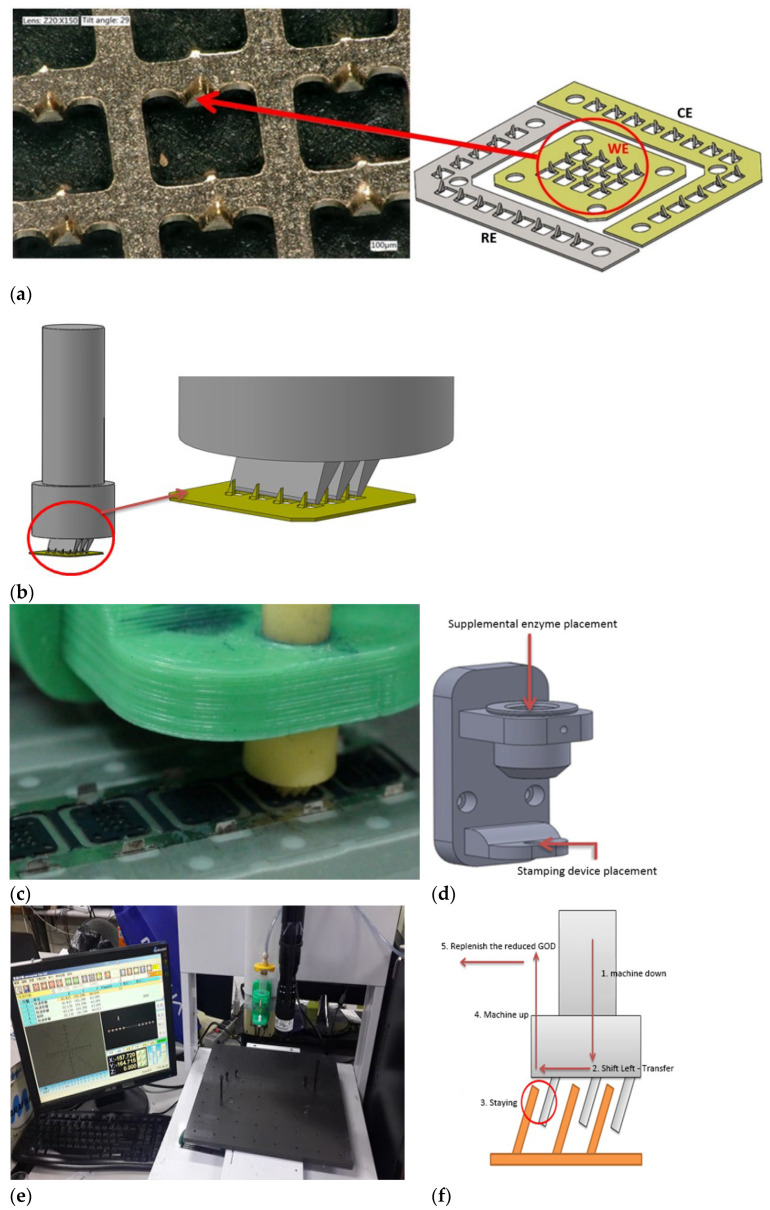
(**a**) Structure of microneedle array sensor; (**b**) schematic diagram of the stamping device and glucose oxidase transfer; (**c**,**d**) stamping device and fixture for the stamping device; all molds are printed on 3D printers at 200 °C and 50 mm/s. (**e**) The Lixi Technology’s 3-axis automatic dispensing machine with molds (**f**) the steps of the transfer process.

**Figure 4 micromachines-13-00478-f004:**
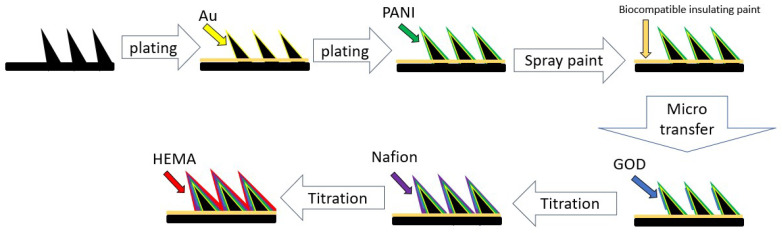
Schematic diagram of the coating; Au and PANI was already electroplated when the microneedle was purchased. Biocompatible insulating varnish is sprayed on before the microtransfer to isolate the signal from the area outside the glucose oxidase coating; two rounds of spraying are sufficient to achieve a good insulating effect. GOD layer: {GOD + PBS} are transferred onto electrodes using micro-transfer method and placed in a constant temperature oven for 10 min at 40 °C and then removed to room temperature. Nafion layer: 10 μL of poly HEMA is titrated onto the microneedle array using a pipette and placed in the dry (approximately 30 min). Poly HEMA layer: 10 μL of poly HEMA is titrated onto the microneedle array using a pipette and placed in the dry (approximately 8 h).

**Figure 5 micromachines-13-00478-f005:**
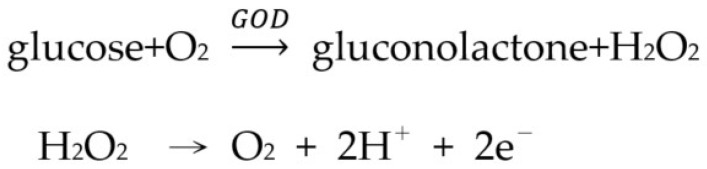
The chemical formula.

**Figure 6 micromachines-13-00478-f006:**
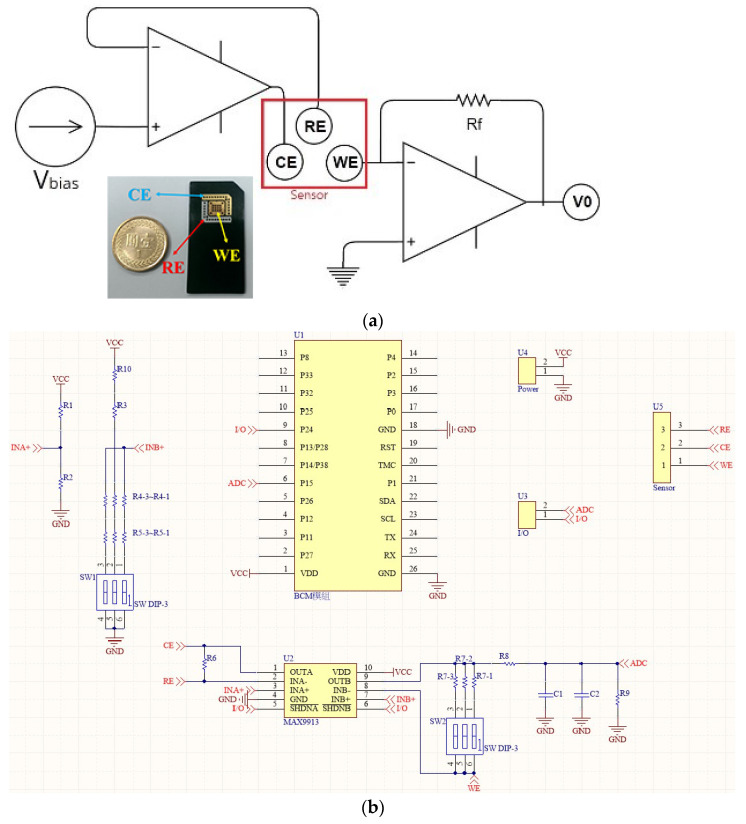
(**a**) Schematic diagram of the sensing circuit; (**b**) design of the sensing circuit.

**Figure 7 micromachines-13-00478-f007:**
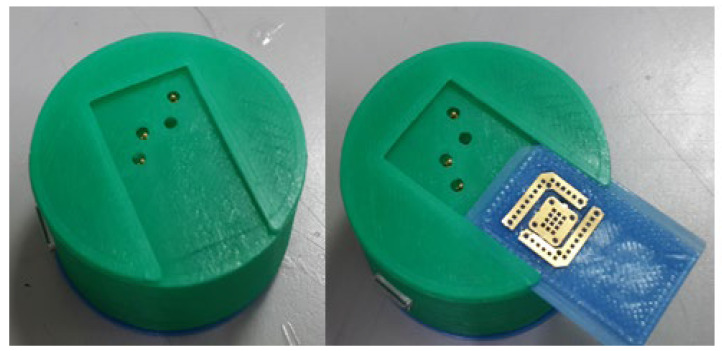
Assembled device and microneedle sensor. The test piece can be changed more rapidly using the spring mechanism.

**Figure 8 micromachines-13-00478-f008:**
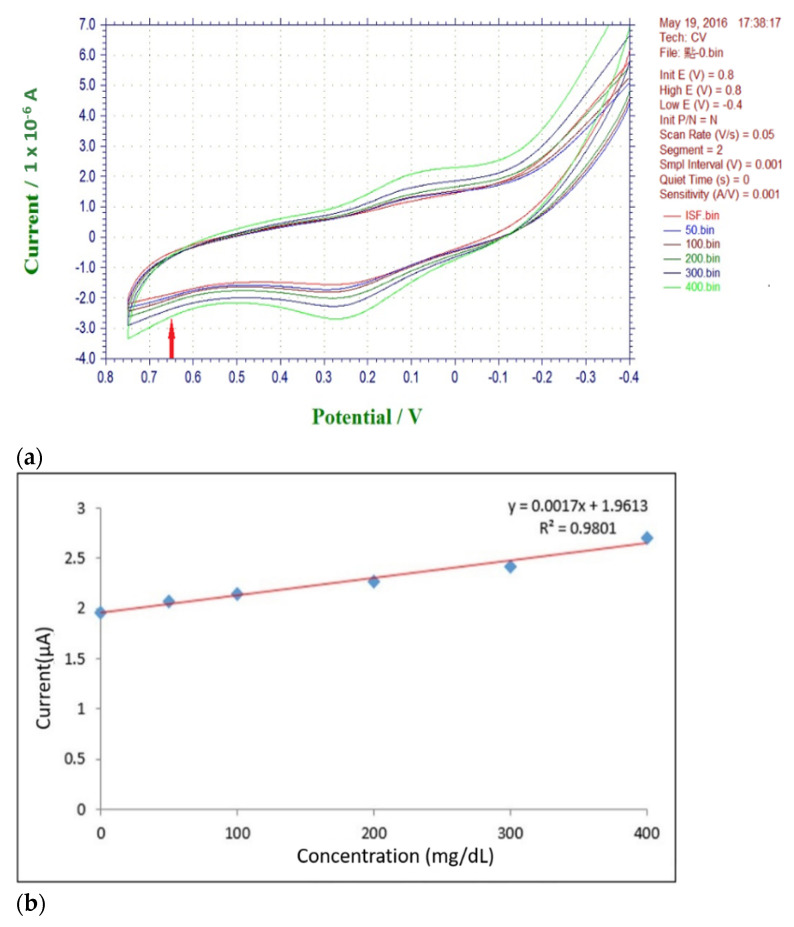
(**a**) Maximum change at 0.65 V for different concentrations of blood glucose; (**b**) current vs. ISF + Glucose concentration at a potential of 0.65 V. Results showed linearity at an operation voltage of 0.65 V.

**Figure 9 micromachines-13-00478-f009:**
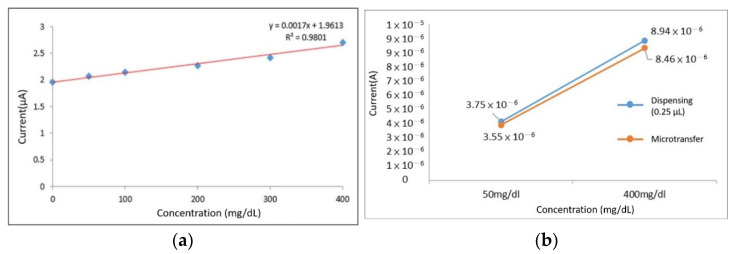
(**a**) Signal difference between a microtransfer and a 5 μL dispense enzyme; (**b**) signal difference between a single microtransfer and 0.25 μL dot enzyme.

**Figure 10 micromachines-13-00478-f010:**
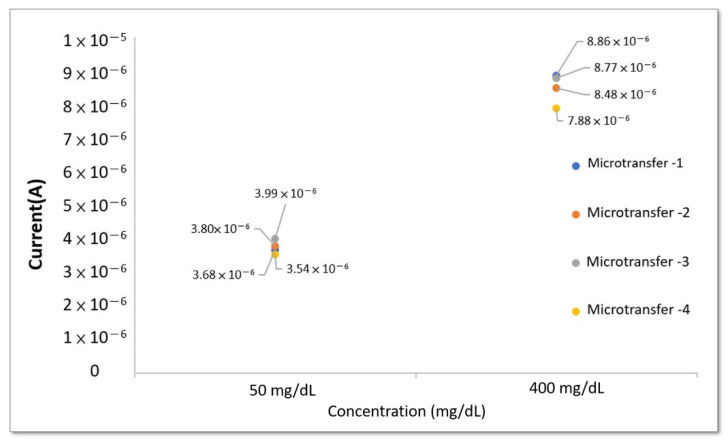
The relationship between current and glucose concentration when the potential is 0.65 V during the continuous transfer of 10 strips of working electrodes.

**Figure 11 micromachines-13-00478-f011:**
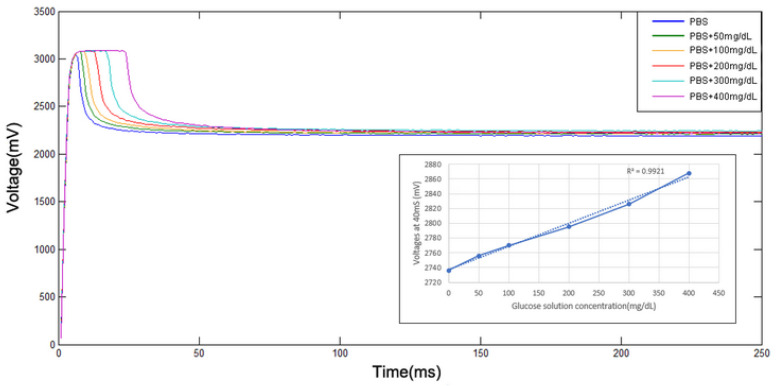
Relationship between current and concentration. In this figure, the sensor shows more significant differences at 40 ms.

**Figure 12 micromachines-13-00478-f012:**
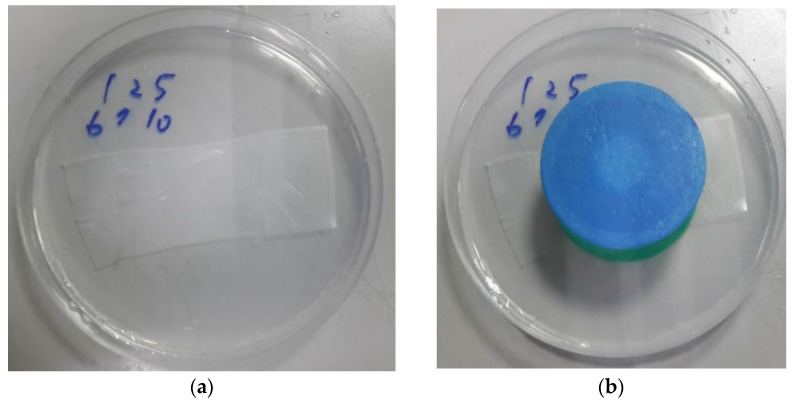
(**a**) agar matrix covered with parafilm and (**b**) CGMS penetration into agar matrix.

**Figure 13 micromachines-13-00478-f013:**
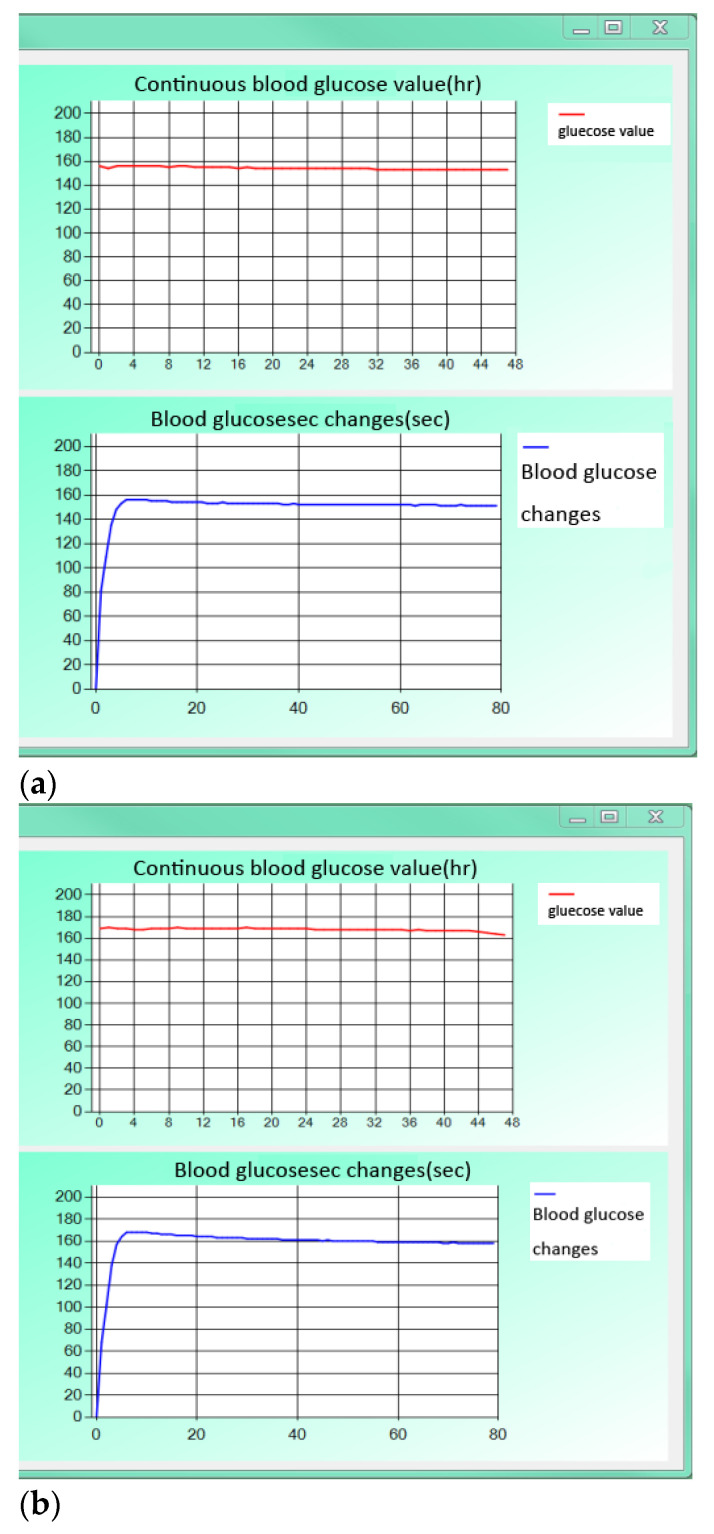
Continuous measurement of concentrations of (**a**) 50 mg/dL; (**b**) 100 mg/dL; (**c**) 200 mg/dL; (**d**) 300 mg/dL; (**e**) 400 mg/dL of glucose solution.

**Figure 14 micromachines-13-00478-f014:**
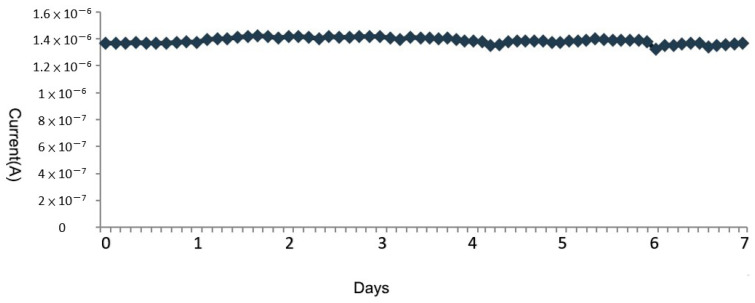
Long-term current change during 7 days.

**Figure 15 micromachines-13-00478-f015:**
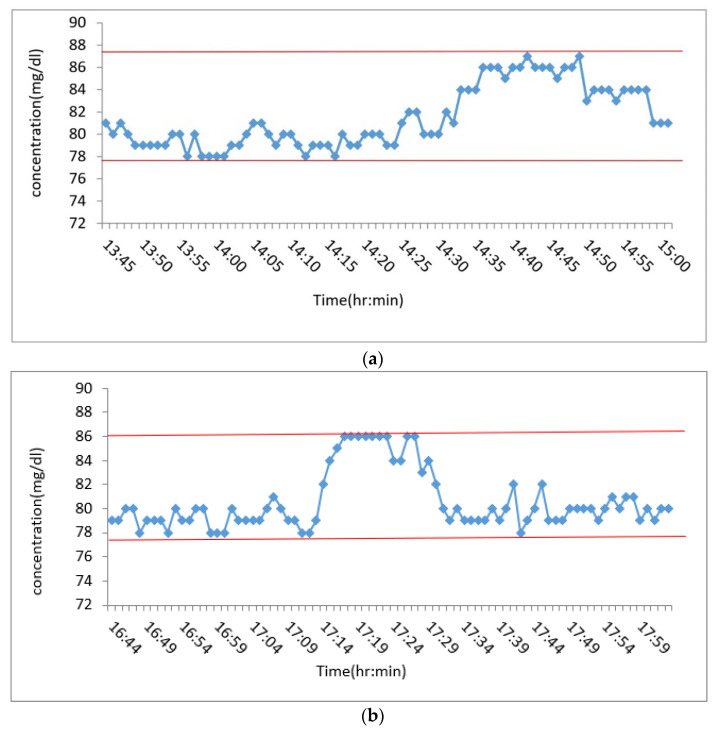
(**a**) Continuous blood glucose signal of subject-1; (**b**) continuous blood glucose signal of subject-2.

**Table 1 micromachines-13-00478-t001:** Characteristics of the commercial continuous glucose monitoring systems.

	Dexcom G5 Mobile CGM System	FreeStyle Navigator	Medtronic iPro 2	Abbott FreeStyle Pro
Target site	Skin	Skin	skin	skin
Sensor lifespan	7 day	5 day	6 day	14 day
Length of microneedle probe	13 mm	6 mm	8.5 mm	5 mm
Sensor warm-up	2 h	10 h	2 h	1 h
Calibration	Every 12 h	Every 12 h	3 on Day 1, 4 per day after	Factory Calibrated

**Table 2 micromachines-13-00478-t002:** Transfer and verification of error transfer.

	Microtransfer	Dispensing (0.25 µL)	Signal Error
50 mg/dL	3.75 × 10^6^	3.55 × 10^6^	5.75%
400 mg/dL	8.94 × 10^6^	8.46 × 10^6^	5.382%

**Table 3 micromachines-13-00478-t003:** Micro-transfer variability test error.

Concentration	Microneedle-Random Sampling						
	Transfer-1	Transfer-2	Transfer-3	Transfer-4	Average	Standard Deviation	Variability Test Error
50 mg/dL	3.68 × 10^6^	3.80 × 10^6^	3.99 × 10^6^	3.54 × 10^6^	3.75 × 10^6^	1.6 × 10^7^	4.266%
400 mg/dL	8.86 × 10^6^	8.48 × 10^6^	8.77 × 10^6^	7.88 × 10^6^	8.50 × 10^6^	4.43 × 10^7^	5.220%

## Data Availability

Not applicable.

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
