# Peer review of "Continuous Glucose Monitoring System Based on Percutaneous Microneedle Array"

_micromachines, 2022, doi:10.3390/mi13030478_

Round 1

Reviewer 1 Report

This paper talked about the development of a less-invasive CGM device. The content covered layout design, board circuit, stability, e-chem and human tests etc. Generally, this paper contains some interesting information and can be useful as a supplement to current CGM community. 

1) Label of Fig. 1 is missing. Numbering of figures in a format like Figure 2.1, 2.2, 2.3 etc is not common, maybe consider correct it back to Figure 1,2,3 etc.

2)Fig. 3.1, What's the total electrode area considered in the signal? why s gold chosen as the electrode material, where typically Pt was used in literature. The baseline current when no analyte presented is 2 uA. is this an expected baseline current level?

3)Fig. 3.5, 100 mg/dL data is missing.

4) The quality of figures are very different from figure to figure. for example, Figure 3.3 looks fine, but Figure 3.5 has very low resolution. 

5) Authors need to add more number of references, especially more up to date papers.

Author Response

1) Label of Fig. 1 is missing. Numbering of figures in a format like Figure 2.1, 2.2, 2.3 etc is not common, maybe consider correct it back to Figure 1,2,3 etc.
Revised: change the Figure number

2)Fig. 3.1, What's the total electrode area considered in the signal? why s gold chosen as the electrode material, where typically Pt was used in literature. The baseline current when no analyte presented is 2 uA. is this an expected baseline current level?
We use Au because it is easier to be electroplated to stainless steel microneedle than Pt. In addition, also, because large reaction area of mcroneedle array and pouros HEMA, a baseline of 2 uA is to be expected.

3)Fig. 3.5, 100 mg/dL data is missing.
Revised, we add the missing data.

4) The quality of figures are very different from figure to figure. for example, Figure 3.3 looks fine, but Figure 3.5 has very low resolution. 
Revised ,the chart are changed

5) Authors need to add more number of references, especially more up to date papers.
Revised, 10 more related and up-to date references added

Reviewer 2 Report

See attachment for my feedback

Author Response

In the manuscript Chien et al. describe the development (and preliminary characterization) of a continuous blood glucose monitoring system, based on a microneedle array onto which enzymes are immobilized. Although rather well-written, the manuscript seriously lacks information. There is no description of how the needle array is designed/fabricated, the circuit and transmission model are ‘posited’ without motivation/explanation, and experiments cannot be repeated/verified by readers due to incomplete descriptions. As a consequence, for the reader it is not possible to repeat the fabrication nor experiments. Moreover, the provided preliminary characterization is by far not convincing, and a subjective judgement of these results is done (i.e. no proper and fair comparison with literature). Overall, the manuscript lacks clarity, is incomplete and not of an acceptable scientific level. Based on the above-mentioned contents-wise aspects, I recommend rejection. Below my comments in detail.

Grammar

ï‚· Abstract:: the abstract should be written in the ‘present tense’ (so, words like ‘included’, ‘was’ and ‘were’ should be changed into ‘include’, ‘is’ and ‘are’).

Revised

ï‚· Lines 43/51: the word ‘scholars’ should be replaced by ‘researchers’.

Revised

ï‚· Lines 60-66: use the ‘present tense’ about own work/investigations.

Revised

ï‚· Line 88: Table 1.1 should be numbered as Table 1.

Revised

ï‚· Line 155-158: the caption contains two identical sentences.

Revised

ï‚· Section 2.2: reagents etc. are ‘purchased’ or ‘bought’, not ‘ordered’.

Revised

ï‚· Section 3.2.2: why is section 3.2.1. not used as number/header?

Revised

ï‚· Fig 3.3: subfigures (a) and (b) are identical… Fig. 3.3(b) appears to be wrong.

Revised

ï‚· Figures: should be numbered in numerical order, i.e. 1, 2, 3 etc. Not 2.1, 2.2 etc.

Revised

ï‚· Figure captions: most captions are not in the caption-format, but in main text-format.

Revised

Contents

ï‚· Line 22: is the indicated ‘1 mm’ the length of the micro-needles in the array? As phrased now, it appears to be the length (or width) of the array… Please clarify. Additionally, if the length of the needles is 1 mm, why are they then classified as microneedles (what is the micro-aspect?)?.

Revised Line 22

Modify the narrative.

Each microneedle is 1 mm in length, 0.25 mm in width and 0.1 mm in thickness. It needs a needle with enough length (in our case 1 mm) to penetrate the skin to reach the ISF in dermis layer without causing any essential trauma. Therefore, the width and thickness of the needle should be around several hundred micrometers. That is why people call this size of needle as microneedle.

ï‚· Line 48: which enzymes may be used/damaged? Please specify.

Revised line 48  

The enzyme is glucose oxidase.

ï‚· Line 57: which protective coatings other than hydrogel can be used? Please specify.

Revised Line 57

The protective coating used in some blood glucose experiments is HEMA.

ï‚· Lines 68-73: in the description of commercial meters 4 types are mentioned, but no details/info on non-invasive and disposable types are provided. Please include (or explain why no info of these 2 types is listed).

Revised Line 68~73

ï‚· Line 72: where does this 15% error in measuring blood glucose come from? Provide (a) reference(s) to evidence this statement.

Revised lline 71

According to ISO15197

ï‚· Line 77/Table 1: are the indicated lengths (6mm-13mm) the needle-lengths? Or of the probe/complete device? Please clarify.

Revised line 77

They are needle-lengths.

ï‚· Lines 74-86: the authors ‘mix’ information about commercial systems with their own CMGS, but this is a not 100% clear. Please include terminology such as “In our CMGS due to the reduced length of the microneedle arrays…” etc. Moreover, provide references to the commercial systems (e.g. websites of the manufacturers).

Revised line 76

Add reference sources about different brands of CGMS.

ï‚· Lines 91-95: please for clarity include a sketch of the conceptual approach of the CGMS, indicating the various elements/components (needle array, circuit module, transmission module etc.).

Revised line 98

ï‚· Line 110: Where does the indicated 6% come from? Why not 3%, or 15%?

Revised 124

In reference 29, 90% of the sensors are within 10% error after production, so we use this value as a reference.

ï‚· Section 2.1/ Lines 110-113: there is no information about the fabrication procedure of the needle array (are the needles make of silicon?), nor any info on geometrical data (needle length, diameter, layout of an array) etc.

Revised 107

Added geometrical data.

ï‚· Lines 117-119: the authors state that ‘if on the stamping devices is fixed, the amount transferred is constant’. However, where can I see/read evidence of these experiments? Also, information on interaction length (between needles and stamp) is not provided, nor the contact time… As it is now, the information is too subjective and cannot be verified by the reader.

Revised line 107

I deleted the description of the transfer amount because there was a problem with that experiment and added the explanation of the microneedle.

ï‚·Line 164: provide references to the ‘polymer deposition method’, as well as experimental details. As it is now, the reader cannot repeat the experiments… the text is too descriptive, rather than scientific.

Revised line 152

Added more scientific explanation about high polymer entrapment method.

ï‚· Lines 166-173: there is no explanation what the working electrode (nor reference and counter electrode) are… presumably the WE is the needle array, but what are the CE and RE? Please explain, including in Fig. 2.3(a).

Revised line106.180

Added more explanation about WE, CE, and RE.

ï‚· Lines 166-173: according to the text 4 layers are deposited, however, in Fig. 2.2 6 layers are mentioned (Au, PAI, paint, GOD, Nafion and HEMA). Please explain the function of all materials, as well as deposition times and layer thicknesses.

Revised line 155~164

Explain more about the function of each layer and the manufacturing method.

ï‚· Section 2.3: a scheme + short explanation on the electrode sensing principle is needed, for the readership of Micromachines.

Revised line 178~182

Added the electrode sensing principle.

ï‚· Line 198: please explain the function/meaning of the indicated module (BCM20737S) and MAX9913, and add the manufacturer. In fact, there is no discussion on the design (why is it like it is), no motivation…

Revised line 204~223

Add motive and introduction.

ï‚· Line 218: what is the manufacturer of the potentiometer ‘CHI’s 1025B’?

Revised line 238

ï‚· Line 226/Fig 3.1: Linearity is best for a control level of 0.65V: I cannot verify this statement, since in Fig. 3.1 only data for 0.65V is shown, not for other potentials… so, this statement is subjective.

Revised line 249

Add more experimental data to explain the selection of reference potential.

ï‚· Line 235: judgement depends on color and depth: is this done by human eye? Is that a fair criterion? Since not all human eyes ‘see’ the same…

This paragraph has been deleted.

ï‚· Line 237: evaporation occurred after a while => what is ‘a while’? Be precise in terms of time.

This paragraph has been deleted.

ï‚· Line 239: the effect was less significant => which effect? Not clear to me…

This paragraph has been deleted.

ï‚· Line 245: transferring dotted glue => not clear what this is, and how it is used.

Revised chapter 3.2.1

I have rewritten this paragraph 255~262

ï‚· Line 248: it is not properly explained how the amount of transferred enzyme can be estimated. Please clarify.

Revised 3.2.1(line261~267)

I have rewritten this paragraph.

I inferred the amount of enzyme transfer by changing the signal size, and then used the inferred amount to test the titration experiment to see if the signal size was the same as the microneedle made by the microtransfer.

ï‚· Line 260: the parameter of time => the time of what/which? Contact stamp-needle array?

Revised: chapter3.2.1(line261~267)

I have rewritten this paragraph

ï‚· Line 263: use of the abbreviation ‘CV’ for cyclic voltammetry as well as coefficient of variation is highly confusing, and should be avoided.

Revised: line 294

ï‚· Figure 3.4: linearity between 2 data-points is trivial, and might be incorrect (no enough data-point to evidence linearity). Please omit such trend-lines.

Revised: line 288

The chart is revised.

ï‚· Table 3.2: the amount of digits for CV% should be 1 (or at most 2).

Revised: line 294

In previous reference papers, a variability error of 5% is acceptable for biosensors

For example: https://journals.sagepub.com/doi/full/10.1177/0954411919883788

ï‚· Lines 279-280: this sentence is not part of the Figure-caption… and how is ‘good resolution’ quantified/evidenced by the authors?

Revised: line 302

the correct current is 40 ms

The chart are changed

ï‚· Figure 3.7: only the graphics in the Figure are of interested (not the layout of the software), and in its current size these graphs/axes-text cannot be read properly.

Revised line 322

ï‚· Fig. 3.8: please merge all data on one time-line plot, not in 2 sub-figs.

Revised line 344

Data has been merged

ï‚· Lines 316-318: due to incorrect grammar, the meaning of this sentence is unclear. Please rephrase.

Revised line350~360

I rewrote this paragraph

ï‚· Lines 324-326: please use 24h-indications in the text, similar to the x-axis of Fig. 3.9. Moreover, it should also be mentioned that these graphs show data of test-person upon eating.

This experiment is not yet complete and the final data will be updated in the next reply

ï‚· Lines 346-358: given the subjective interpretation of presented preliminary data (a comparison with literature data is fully lacking: is this CGMS better then commercial systems?! I am not convinced…) these lines are too speculative/optimistic, and out-of-proportion. Please omit.

Revised line371~379

I rewrote this chapter

Round 2

Reviewer 1 Report

Authors did good modifications after 1st review, the manuscript looks better.

If editor and other reviewers decided to accept, I'm ok with it.

Reviewer 2 Report

The authors have properly addressed my comments, and adapted the manuscript accordingly. Provided some (small) additional modificiations, the paper can be accepted for Micromachines.

  • Line 94: microneele => microneedle
  • Figure/Table-captions: not in the template-format/style (see template!)
  • sometimes articles ('the'/'a'/'an') are missing, e.g. line 131 (... the contact...)
  • lines 159-161 are in a different fonttype
  • it should be μL, not uL (use of Greek symbol, e.g. in lines 261/264)
  • Fig. 13 can be plotted in 2x2 matrix (a and b in 1st row, b and c in 2nd row, e in 3rd row)
  • the 'p.m.' in lines 355-357 is redundant (24h notation is used)